# Resilience, Stress, and Cortisol Predict Cognitive Performance in Older Adults

**DOI:** 10.3390/healthcare11081072

**Published:** 2023-04-09

**Authors:** Noelia Saez-Sanz, Isabel Peralta-Ramirez, Raquel Gonzalez-Perez, Enrique Vazquez-Justo, Alfonso Caracuel

**Affiliations:** 1Mind, Brain, and Behavior Research Center (CIMCYC), University of Granada, 18071 Granada, Spain; 2Department of Psychology and Anthropology, University of Extremadura, 06006 Badajoz, Spain; 3Department of Personality, Evaluation, and Psychological Treatment, University of Granada, 18071 Granada, Spain; 4Department of Pharmacology, Centro de Investigación Biomédica en Red de Enfermedades Hepáticas y Digestivas (CIBERehd), School of Pharmacy, Instituto de Investigación Biosanitaria ibs.GRANADA, University of Granada, 18012 Granada, Spain; 5Center for Research, Development and Innovation (CIDI-IEES), European Institute for Higher Studies, 4824-909 Fafe, Portugal; 6Department of Developmental and Educational Psychology, University of Granada, 18071 Granada, Spain

**Keywords:** stress, cognition, older adults, resilience, cortisol, stressful life events

## Abstract

Objective: to determine the relationship between stress, resilience, and cognitive performance in older people without dementia. Method: multiple linear regressions were performed using measures of cognitive performance as dependent variables, and measures of stress and resilience as predictors in a sample of 63 Spanish elderly people. Results: participants reported low levels of stress during their lifetime. In addition to socio-demographic variables, greater stress was related to better delayed recall and worse letter–number sequencing and block design. Higher capillary cortisol was associated with lower flexibility on the Stroop task. Regarding protective factors, we found that greater psychological resilience was related to higher scores on the Addenbrooke’s Cognitive Examination-III, letter–number sequencing, and verbal fluency. Conclusion: in older people with low stress, apart from age, gender, and education, psychological resilience is a significant predictor of global cognitive status, working memory, and fluency. Likewise, stress is related to verbal memory functioning, working memory, and visuoconstructive abilities. Capillary cortisol level predicts cognitive flexibility. These findings may help to identify risk and protective factors for cognitive decline in older people. Training-based programs to reduce stress and increase psychological resilience may play an important role in preventing cognitive decline.

## 1. Introduction

Population aging is a growing global phenomenon, as indicated by the projection that there will be 1.5 billion older people by 2050 [1]. The prediction that 75 million will have dementia [2] highlights the relevance of taking action against cognitive decline. The risk factors associated with cognitive decline are very diverse. First, advanced chronological age and low level of education are the most predominant [3], while other risk factors include lifestyle and health, such as hypertension, smoking, obesity, sedentary lifestyles, diabetes, excessive alcohol consumption, stroke, and traumatic brain injury [4]. Within the emotional domain, the main risk factors are stress and depression and anxiety [5,6]. Among the protective factors against cognitive decline, physical activity [7] and cognitive reserve [8] play a key role. Cognitive reserve includes education as a protective factor, and refers to the adaptability of cognitive processes that help to explain the differential susceptibility of cognitive abilities, or day-to-day function to aging or brain pathology [9]. Individual differences in cognitive reserve are due to both innate factors (such as brain volume) and acquired factors [10], most notably educational level [11]. The resulting balance between risk and protection determines the onset of cognitive impairment symptoms [12]. Concerning psychological stress as a risk factor, a review by Stuart and Padgett [6] concluded that there is sufficient evidence to suggest an association between this construct and dementia. Although this conclusion should be taken with caution, because of the heterogeneity of the studies reviewed, it provides an incentive to investigate whether psychological stress is associated with the less severe cognitive impairment that affect many older people without dementia.

In the general population, psychological stress arises when the individual’s relationship with their environment is perceived as threatening and beyond their capacity to cope, which endangers their well-being [13]. In many cases, the event perceived as threatening is a stressful life event (SLE) [14]. The physiological correlate of the acute psychological stress experienced following such an event includes the activation of the hypothalamus–pituitary–adrenal (HPA) axis, and the secretion of several glucocorticoids, among which cortisol plays a major role [15]. When stress levels remain high on a daily or chronic basis following the acute event, the regulation of the HPA axis is altered, and the increase in glucocorticoids is maintained, altering the basal activity of the amygdala, the hippocampus, and the medial prefrontal cortex. As a consequence of this brain dysfunction, cognitive function may be altered [16].

Findings from studies using behavioral measures to explore the relationship between SLEs and cognitive status in older people are inconclusive. While Zuelsdorff et al. [17] found that higher stress scores were associated with poorer cognitive performance in later life, specifically in processing speed and flexibility, previous studies have found results to the contrary. Thus, Grimby and Berg [18] found no relationship between SLEs and cognitive decline, except in bereavement situations that were related to decline, and Rosnick et al. [19] found that certain SLEs experienced by older people in the past year (e.g., death of a friend) were associated with higher cognitive performance. In a longitudinal study with a three-year follow-up of people with mild cognitive impairment (MCI), they found that in addition to the number of SLEs, the perceived severity of SVEs was associated with the transition from an MCI diagnosis to dementia, while participants with normal cognitive performance did not progress to MCI [20].

In studies using cortisol as a psychophysiological measure of stress in older people, various findings seem to suggest that the type of measure used could play a key role. Those who have measured mean daily salivary cortisol level, such as Peavy’s team, found that, in a longitudinal study, a lower level of cortisol awakening response (CAR) acted as a predictor of moving from normal cognitive status to a diagnosis of MCI, but not in the change from MCI to dementia [21]. However, Popp et al. [22] found differences between the change from MCI to Alzheimer’s disease (AD) when measuring cortisol levels in cerebrospinal fluid. Findings using plasma cortisol levels also showed a similar (albeit nonsignificant) trend. Other studies of plasma cortisol in people with AD found a relationship between high cortisol levels, the speed at which their symptoms worsened, and their scores on cognitive functions associated with the temporal lobe, primarily memory [23]. Differences in the fluid from which the cortisol level is extracted, and in the cross-sectional or longitudinal nature of the measurement, make it difficult to compare results. Cortisol levels measured in plasma or saliva samples only reflect concentrations within a very recent time window (minutes). Samples aimed at capturing acute cortisol changes occurring within minutes (saliva, plasma) or hours (urine) are less useful for reflecting overall basal levels, as they only collect information at a specific point in time [24]. Capillary cortisol concentrations may instead be a more appropriate measure of chronic HPA axis activation in the preceding months [25]. To our knowledge, only one study has measured both capillary and saliva levels to determine the relationship between cortisol and cognitive performance in older people. This study found that higher salivary cortisol levels were associated with poorer attention and short-term verbal memory. In contrast, lower hair cortisol levels were associated with poorer working memory, learning, and verbal memory [26].

Among the protective factors against cognitive decline, we did not find strong evidence for any related to stress management. However, in studies of stress in other health domains, we found that psychological resilience acts as a buffer against the effects of stress. One review study has revealed that high psychological resilience in older people is associated with benefits such as the reduced risk of depression and mortality, improved self-perception of successful aging, higher quality of life, and improved lifestyle behaviors [27]. According to the American Psychological Association, psychological resilience is the process of adapting well to or recovering from adversity, trauma, tragedy, threat, or significant sources of stress [28]. Therefore, to study the effect of stress on the global cognition of older people, it would be necessary to include the variable psychological resilience. However, to date, no studies have done so. Only one study has attempted to determine whether there is a direct relationship between psychological resilience and cognitive status in older people, in this case with a sample of adults up to 73 years old with HIV (range 40–73 years; 61% over 50 years old). The results indicated that more resilient people scored higher on working memory verbal fluency, executive functioning, information processing speed, learning, and global cognition, but not on delayed recall and psychomotor skills [29]. It is noteworthy that a very recent study has found in the elderly a relationship between psychological resilience and a specific domain of cognitive performance, working memory [30].

The findings in the literature on the association between stress and cognitive decline in older people suggest the need to determine the role of SLEs, cortisol, and psychological resilience. Thus, our study objective was to determine the relationship between the cognitive status of older people without dementia, and stress and psychological resilience.

## 2. Method

### 2.1. Participants

The sample consisted of 63 participants (69.8% female) aged between 58 and 93 years (M = 76.49 years, SD = 8.35), and a mean education of 8.70 years (SD = 4.02). The sample size calculation was based on total verbal learning, the only cognitive variable shared with the only study conducted with hair cortisol in older people. The effect size for verbal learning in the multiple regression of Pulopulos et al. [26] was 0.44. With these data, considering the multiple linear regression and the parameters of two tails, six independent variables, an alpha level of 0.05, and a power of 0.95, the resulting sample size calculated with G-Power v3.1.9.7 (Heinrich-Heine-Universit, Düsseldorf, Germany) [31] was 48 participants. However, we chose to include an additional 15 participants to meet the recommended minimum of 10 participants for each variable in the regression analysis.

Participants were recruited through social media and community centers in the metropolitan area of Granada, Spain. Inclusion criteria for the study were: (i) being over 55 years of age; (ii) having at least a basic level of literacy; (iii) and having an MMSE score ≥ 21, as recommended by MacKenzie et al. [32]. Exclusion criteria included suffering from a significant medical disorder (e.g., insulin-dependent diabetes, chronic inflammation), having been diagnosed with dementia or a major mental disorder (e.g., depression, post-traumatic stress disorder), or receiving corticosteroid treatment.

Participation was voluntary, and each participant read and signed an informed written consent document. All procedures performed in this study were in accordance with the ethical standards of the institutional research committee, and with the 1964 Helsinki declaration and its later amendments. This study was approved by the Human Ethics Research Committee of the University of Granada (235/CEIH/2016).

### 2.2. Instruments

#### 2.2.1. Tests to Assess Cognitive Status

Mini-Mental State Examination (MMSE) [33]. It is the most widely used instrument to measure global cognitive performance. Cronbach’s alpha is 0.91 [34]. The global score has been used.Addenbrooke’s Cognitive Examination 3rd version (ACE-III) [35]. This is a brief cognitive assessment test that includes attention, memory, language, verbal fluency, and visuospatial skills. The global score has been used. Cronbach’s alpha is 0.927 [36].Hopkins Verbal Learning Test-Revised (HVLT-R) [37]. It contains twelve nouns, four words each from three semantic categories, to be learned over the course of three learning trials, and 25 min later, a delayed recall trial and a recognition trial are completed. The sum of the number of correct words on the three learning trials was used as a learning index, and the number of words on the delayed recall were included as the memory index. Both indexes achieved the better reliability test–retest results in the original study (0.74 and 0.66 respectively). There are no studies on the internal consistency of the test.Letter–Number sequencing subtest of the WAIS-III [38]. This task measures the ability of short-term memory to process and sequence information. It consists of listening to a series of letters and digits and then reporting the stimuli with the letters in alphabetical order, and the digits in ascending numerical order. The series will become more difficult as the subject becomes more successful. Overall number of correct answers was considered for scoring. For people between 55 and 89 years old, Cronbach’s alpha is between 0.85 and 0.99 [39].The Block design subtest of the WAIS-III [38]. It is primarily a measure of visuospatial abilities. The participant is presented with blocks with solid red surfaces, solid white surfaces, and surfaces that are half red and half white. The number of blocks is progressively increased to a maximum of nine to reproduce an increasingly difficult pattern. Overall number of correct patterns was considered for scoring. For people between 55 and 89 years old, Cronbach’s alpha is between 0.85 and 0.99 [39].The FAS test of the Neurosensory Center Comprehensive Examination for Aphasia (NCCEA) [40]. This task measures fluency with the phonemes F, A, and S. Listening to each of these phonemes, the participant must say as many words as possible initiated with each of these sounds in 60 s. The total number of correct words was considered for scoring. Cronbach’s alpha is 0.83 [41].Animals and Fruits Naming [42]. This is a semantic fluency task in which the participant is asked to name in 60 s all the words belonging to each of the two categories, first animals and then fruits. The total number of correct words was considered for scoring. There are no studies on the internal consistency and test–retest reliability for older people is 0.70 [43].The Stroop task of the Delis–Kaplan Executive Function System (D–KEFS) [44]. It is a measure of cognitive flexibility, selective attention, cognitive inhibition, and information processing speed. The three main indices used were: inhibition (switching vs. combined naming and reading; calculated with the formula time spent on Part 4 minus the total time spent on Parts 1 and 2), interference (inhibition vs. color naming; time spent on Part 3 minus time spent on Part 1), and flexibility (inhibition/switching vs. inhibition; time spent on Part 4 minus time spent on Part 3). In all three indices, the measure used was the total time taken to perform the test. Therefore, a higher time score would indicate worse performance. For people between 50 and 89 years old, Cronbach’s alpha ranges between 0.77 for the oldest and 0.86 for the youngest.

#### 2.2.2. Stress Testing

The Connor and Davidson Resilience Scale (CD-RISC) [45]. This scale measures psychological resilience based on personal competence, standards, and tenacity; trust in one’s instincts; tolerance of negative affect and resilience to the effects of stress; acceptance of change; secure relationships, degree of control; and spiritual influences. In this study, we used the total global score. The higher the score, the higher the degree of resilience, ranging from 0 to 100. Cronbach’s alpha is 0.08 [46].Stressful Events in the Elderly Scale (EAE) [47]. This scale was developed in the Spanish population to measure current and lifetime stress in older adults. The participant is asked whether 51 stressful life events (SLEs) have occurred in their lifetime in the areas of physical and mental health, and social and economic aspects. These include everyday stressful life events (e.g., living together, health problems, dependency, or financial situation) and other more specific events, such as losing someone close. For each stressor experienced, participants have to answer: (1) “with what level of intensity did it affect you in the past (past stress)?” and (2) “is it still affecting you in the present (current stress)?”. The score used is the sum of the intensity of stress. Cronbach’s alpha is 0.81.Hair cortisol test. The samples were analyzed at the Department of Pharmacology of the University of Granada, Spain. The method of obtaining and processing hair to determine cortisol levels in the last three months is described in previous studies [48,49,50].

### 2.3. Procedure

Recruitment was carried out through the distribution lists of the University and the Granada City Council, as well as through direct dissemination to the professionals of the community centers for the elderly in the metropolitan area of Granada. The evaluation was carried out between March 2019 and February 2020 in the community centers, individually and in three sessions on different days, each lasting about 90 min with a 25 min break. In the first session, the informed consent, socio-demographic questionnaire, ACE-III, FAS, and animals and fruits naming tests were administered. In the second session, HVLT-R, CD-RISC, and D–KEFS tests were administered. In the third session, letter–number sequencing, block design, EAE, and hair sampling tests were administered. The tests were administered by a trained psychologist.

### 2.4. Data Analysis

As the sample population had a higher percentage of women, we checked for possible gender differences in the cognitive variables. All except fruits and animals (t = −2.970; *p* = 0.004) were the same for females and males. This effect was controlled for by including sex as a factor in the regression analyses. Multiple linear regression analyses were conducted for exploratory purposes to determine whether measures of stress and resilience (independent variables) predicted global cognitive performance and its different domains (dependent variables). In all analyses, age, years of education [3], and gender [51] were included, to control for the effect of these factors on cognition. The other predictors included were stress intensity, capillary cortisol concentration in the three months prior to the assessment, and psychological resilience. To ensure that the model estimates were not unstable due to multicollinearity, we conducted collinearity diagnostics on each set of variables entered into a model, using the variance inflation factor (VIF), tolerance, condition indices, and variance percentages. It is conventional to regard with suspicion any variable with a VIF greater than 10 [52]. In addition, the Durbin–Watson test was carried out to establish error independence [53], and to confirm that these were within the range of 1.5 and 2.5 [54]. The results were found to be independent and within this range.

Data were analyzed with the statistical package SPSS Statistics 22 [55], and are openly available at https://osf.io/gup7h/ (accessed on 17 January 2023).

## 3. Results

Table 1 shows descriptive statistics of socio-demographics, cognition, and stress variables included in the regression models. The mean global cognitive performance measured with the ACE-III is a direct score of 76.97, and the mean of the percentiles of the participants concerning their reference group [3] places the sample at a mean of 38.06 (SD = 24.45). Regarding the intensity of the SLEs, 61.9% of the sample had a low stress level (percentile < 16), 36.5% a medium level, and only one person had a stress level above the 84th percentile [47].

All multiple linear regression analyses were significant and met the principles of noncollinearity and independence of errors. Of the three socio-demographic variables, age is a significant predictor for all cognitive variables, except flexibility. Years of education is a predictor of ACE-III, WAIS-III (letter–number sequencing and block design), FAS test, and flexibility scores, although the latter only marginally. Gender is only a significant predictor of animals and fruits naming (see Table 2).

The intensity of experienced stress predicts delayed recall, letter–number sequencing and block design. Resilience predicts ACE-III, letter–number sequencing, animals and fruits naming, FAS test, and block design scores, although the latter only marginally. 

## 4. Discussion

This study aimed to determine, after controlling for age, years of schooling [3], and gender [51], the relationship between the cognitive status of older people and their level of stress experienced throughout life, hair cortisol, and psychological resilience.

Psychological resilience stands out as the best predictor of overall cognitive performance, as it contributes to relevant domains such as working memory, verbal fluency, and visuospatial abilities. Fazeli et al. [29] found similar relationships, but also with other cognitive functions, such as learning and executive functioning, that have not emerged in our study. The reason for this discrepancy could be due to the fact these are not entirely comparable studies. In particular, their sample consisted of people with HIV and, together with the elderly sample, also included young people from 40 years of age, two variables that are highly correlated with cognitive performance [56,57]. One recent study has also shown an association between resilience and working memory in elderly people [30]. These findings are relevant when analyzed from the point of view of their potential implications for the evolution and intervention of cognitive impairment. In this regard, there is evidence of an association between high verbal fluency scores and a decreased risk of dementia, including the progression from mild cognitive impairment to dementia [58]. In addition, according to the review by Bastin and Salmon [59], semantic fluency has been identified as the best cognitive marker of Alzheimer’s disease, ahead of phonological fluency. There is also evidence that working memory deficits are associated with greater progression to Alzheimer’s disease in people with mild cognitive impairment [60]. In this respect, this finding on the relationship between psychological resilience and working memory can be framed within the hypothesis that psychological resilience could be a socio-cultural index of cognitive reserve and, as such, would act as a protective factor for Alzheimer’s disease [9]. It would therefore be desirable to continue to study this construct in older people, observe whether the relationship is confirmed, and determine the role it may play in cognitive performance and its evolution.

Regarding stress, we found an association with delayed verbal recall, working memory, and visuospatial skills. These findings might shed some light on the scarce and inconsistent previous findings on stressful life events and their relationship with cognitive performance in older people [17,18,19,20]. In our sample, people with more stress have better verbal memory. To explain these results, we must consider the low average intensity of stress in the sample. Because of this, these results could be explained based on the Yerkes–Dodson law, which posits that low or moderate stress doses can benefit cognitive performance [61]. Nevertheless, the relationship between working memory and experienced stressful events has not been found in other studies [17], so further research is needed.

On the other hand, our findings have revealed that higher psychological stress is associated with worse working memory and visuoconstructive ability. Chen et al. [62] also found this association between stress and working memory in a sample of Chinese-American older people. These findings are in line with classic studies reporting that elevated stress levels impair hippocampal functioning, weakening or disrupting spatial and explicit memory processes served by this structure [63]. Similarly, we found that higher capillary cortisol, the physiological measure of stress in the last three months, is associated with lower cognitive flexibility. However, Pulopulos et al. [26] found the opposite, that lower capillary cortisol was associated with poorer performance in domains such as working memory, learning, and verbal memory. These contradictory findings raise the need for further research in the field, based on capillary cortisol as a longitudinal measure of stress that more adequately reflects the influence of stressful events in real life. There are other studies using measures of cortisol in older people, but they are not comparable to ours because they measured cortisol from saliva or plasma samples, which only reflect immediate time frames of minutes or hours, and are thus of little use in determining the association between long-term endocrine production and cognition.

The findings of the study on psychological resilience could have theoretical implications at the neuroanatomical level. In that sense, the results found show that performance in verbal fluency and working memory has a positive association with resilience. In both cognitive functions, the prefrontal cortex [64] plays a relevant role. Regarding the brain circuits involved in resilience, although studies are still scarce, a recent review places the prefrontal cortex as the key area of resilience in young people [65]. The functional anatomical relationship between cognition and resilience could also be studied longitudinally in the elderly. At the clinical level, both verbal fluency [58] and working memory [66] are predictors of cognitive impairment. Their relationship with resilience allows exploring options for the prevention of deterioration through programs to strengthen this skill. These findings may help to identify risk or protective factors for cognitive decline in older people. It has been shown that both mood [67,68] and stress [6] have an impact on cognitive decline in older people. However, if these results are confirmed, we should not only address the influence of mood and stress, but also consider that the method of coping with adversity could be an important factor in cognitive performance in older people. This possibility could have important clinical implications that would require effective intervention programs that address psychological resilience in older people [68], which may serve as a protective factor for cognitive performance. In fact, programs to improve psychological resilience in older people are already being conducted, with good results in achieving greater self-efficacy [69], and improving stress coping skills and daily functioning [70]. It would be advisable that future lines of research focus on the impact that these programs may have on the cognitive performance of older people.

The limitations of our study include the sample size, which, although it meets the recommended minimum of 10 participants for each factor included in the regression models, means that the findings should be interpreted with caution [71]. With a larger sample size, structural equation models could be utilized to determine the relationships between cognitive performance and the psychological stress factors associated with stressful life events, cortisol as a physiological correlate, and psychological resilience. Another limitation of our study is the cross-sectional design, as this does not allow us to establish how cognitive performance will evolve according to the predictors found.

## 5. Conclusions

In conclusion, in older people with a low level of stress and without dementia, psychological resilience is the best predictor of global cognitive state, working memory, and fluency. The level of stress is related to delayed verbal recall, working memory, and visuospatial ability. Therefore, this study demonstrates the special relevance of our ability to adapt to adversity or significant sources of stress as a protective factor in the cognitive performance of the elderly, an aspect that has received relatively little attention to date. 

## Figures and Tables

**Table 1 healthcare-11-01072-t001:** Descriptive statistics of socio-demographic, cognition, and stress variables.

Test	Variable	Mean (SD)	Range
	Age	76.49 (8.35)	58–93
	Years of education	8.70 (4.02)	1–14
	MMSE	27.30 (2.42)	21–30
Cortisol	Hair cortisol (percentiles)	39.35 (34.57)	0.20–100
EAE	The intensity of stress	28.68 (15.93)	0–110
CD-RISC	Overall score	70.59 (16.64)	17–97
ACE-III	Overall score	76.97 (11.86)	48–97
HVLT-R	Total recall score	17.02 (5.75)	3–33
	Delayed recall	5.02 (2.80)	0–12
WAIS-III	Letter–number sequencing	12.11 (5.70)	0–27
	Block design	27.18 (13.08)	0–60
FAS test	Phonemic fluency	27.11 (12.06)	6–56
Animals and Fruits	Semantic fluency	25.67 (7.52)	8–49
D–KEFS	Stroop-interference index	59.22 (49.61)	10–235
	Stroop-inhibition index	51.53 (78.26)	−47–312
	Stroop-flexibility index	26.68 (54.80)	−102–205

MMSE: Mini-Mental State Examination; EAE: Stressful Events in the Elderly Scale; CD-RISC: The Connor–Davidson Resilience Scale; ACE-III: Addenbrooke’s Cognitive Examination 3rd Version; HVLT-R: Hopkins Verbal Learning Test-Revised; WAIS-III: Wechsler Adult Intelligence Scale 3rd Edition; FAS test: The FAS test of the Neurosensory Center Comprehensive Examination for Aphasia; Animals and Fruits: The Animals and Fruits naming test; D–KEFS: Delis–Kaplan Executive Function System.

**Table 2 healthcare-11-01072-t002:** Results of the multiple regression models.

Tests	Dependent Variable	Age *p*-Value	Edu. *p*-Value	Gender *p*-Value	EAE *p*-Value	Hair Cortisol *p*-Value	Resilience *p*-Value	Full Model R² Adjusted (*p*-Value)	Sign. Contrib.	Stand. β
ACE-III	Overall score	<0.001	<0.001	0.075	0.661	0.580	0.001	0.595 (<0.001)	Age Edu. Resilience	−0.513 0.475 0.283
HVLT-R	Learning	<0.001	0.443	0.224	0.097	0.490	0.128	0.373 (<0.001)	Age	−0.537
	Delayed recall	<0.001	0.724	0.931	0.017	0.424	0.345	0.332 (<0.001)	Age EAE	−0.504 0.271
WAIS-III	Letter–number sequencing	<0.001	0.010	0.739	0.037	0.975	0.001	0.415 (<0.001)	Age Edu. EAE Resilience	−0.541 0.273 −0.228 0.342
	Block design	<0.001	0.009	0.329	0.026	0.577	0.057	0.481 (<0.001)	Age Edu. EAE	−0.630 0.265 −0.268
Animals and Fruits	Semantic fluency	<0.001	0.267	0.021	0.223	0.291	0.001	0.354 (<0.001)	Age Gender Resilience	−0.390 −0.262 0.351
FAS	Phonemic fluency	<0.001	<0.001	0.131	0.436	0.572	0.001	0.524 (<0.001)	Age Edu. Resilience	−0.374 0.517 0.311
D–KEFS	Stroop-interference	<0.001	0.949	0.347	0.995	0.441	0.569	0.300 (<0.001)	Age	0.639
	Stroop-inhibition	<0.001	0.247	0.584	0.987	0.387	0.946	0.239 (0.002)	Age	0.521
	Stroop-flexibility	0.115	0.051	0.861	0.887	0.033	0.635	0.129 (0.036)	Hair Cortisol	0.271

Edu: years of education; Resilience: CD-RISC: the Connor–Davidson Resilience Scale; EAE: intensity of events of the EAE Stressful Events in the Elderly Scale; ACE-III: Addenbrooke’s Cognitive Examination 3rd Version; HVLT-R: Hopkins Verbal Learning Test-Revised; WAIS-III: Wechsler Adult Intelligence Scale 3rd Edition; Animals and Fruits: The Animals and Fruits naming test; FAS: The FAS test of the Neurosensory Center Comprehensive Examination for Aphasia; D–KEFS: Delis–Kaplan Executive Function Systems; Stand. β: standardized β; Sign. Contrib.; significant contributors.

## Data Availability

The data can be found on the following website: https://osf.io/gup7h/ (accessed on 17 January 2023).

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
