# Peer review of "Resilience, Stress, and Cortisol Predict Cognitive Performance in Older Adults"

_healthcare, 2023, doi:10.3390/healthcare11081072_

Round 1

Reviewer 1 Report

Review of healthcare-2198870

The objective of the present study was to determine the relationship between protective and risk factors for stress and cognitive performance in an elderly population. Results showed interesting correlations between past and present stress and low punctuation in several cognitive tests. On the contrary, subjects with low levels of stress showed better punctuation in certain psychological and cognitive domains.

Although the objective of the study seems interesting, it is necessary to consider some points for its improvement:

Method

Participants: On the one hand, I think the sample recruited is small. This topic is developed in the last section of the article, where the authors refer it as a negative point of their work.

On the other hand, I think that the sample should be balanced by sex. I calculated that there were recruited 43-44 women and 20 men. Below I indicate a few reasons, including the doi of some articles where it is declared what I indicate:

1. Being a woman is much more likely to suffer stress (10.2174/138161210791792831; 10.1016/j.maturitas.2016.09.004).

2. There are many cognitive abilities related to spatial memory that are determined by sex. For example, men are better at visual-spatial memory and use navigation strategies that are more effective than those often used by women. In addition, they are better at mental rotation of spaces. Conversely, women are more skilled at memory for locations. On this point, I think we should be very cautious about the results obtained in the Addenbrooke's Cognitive Examination 3rd version-ACE-III test. This test includes within the category "visuospatial memory" a series of items that really evaluate perception, visoconstruction and management of focused attention (line 151). I think it is important to stress this because, as I said above, it is known that women show worse performance in visuospatial working memory tasks compared to men (10.1159/000443174; 10.3758/bf03210836).

3. Although they are not conclusive, there are also many studios that confirm a better performance of women in certain verbal fluency tasks (10.5709/acp-0238-x; 10.1177/17456916221082116).

Procedure: It is no explained how the tests were applied. Were they applied one by one or in groups? How were the rooms? Which was the order of application of the different questionaries? This is important to avoid a declain of the attentional level and fatigue (moreover in older population). How many examinators were in there?

I think three 90-minute sessions in one day is mentally very hard. Neuropsychologically speaking, it is not appropriate to make such extensive assessments. Divide the sessions into three days would have been more appropriate. Also, the reduction of time of each session, making a more appropriate selection of the tests. They probably suffered from cognitive fatigue and, therefore, the results cannot be considered free of bias.

Discussion

Line 283. “The findings of this study on the relationship between psychological resilience and global cognition, working memory, and verbal fluency”. I think that it is important to highlight and briefly discuss that all of these cognitive functions are related to the prefrontal lobe and, possibly, to the cognitive reserve (10.1080/09084282.2011.595458).

I think that you will improve your discussion section if you add information about the anatomy associated, cognitive functions and changes that occur during aging. The discussion is very focused on talking about correlations and looking for examples from other studies with similar results. I think it is more interesting to discuss why these correlations can occur.

I think highlighting whether the correlation is direct or inverse is not really what is interesting about your article. What is important is the implication of these results in the population that has suffered or is suffering from stress. That is why I suggest ordering the information by protective factors and risk factors (as you did in the introduction section) and not by direct and inverse correlations. In this regard, the title of the paper and the abstract would also be changed.

References

I understand that you need to reference old studies because few have analyzed these specific data, but I believe that using more up-to-date literature would improve quality.

Author Response

Estimado revisor,

Consulte el archivo adjunto,

Atentamente,

noelia

Reviewer 2 Report

Dear authors, 

The article by title: 

 Stress and Cognitive Performance in Older Adults: Direct and Inverse Relationships with Stressful Life Events, Cortisol, and Resilience.

It aims to determine the relationship between protective and risk factors for stress and cognitive performance in older people without dementia.

I am going to make some suggestions for improvement below, with the aim of improving its visibility, citations, downloads and internationalisation.

In the introduction:

After the third line of your introduction I suggest you incorporate these references to increase the internationalisation of your study, citations and downloads:

 "The practice of physical activity is one of the variables that we should not forget in this area, as it means an improvement in the quality of life of older people as well as a guarantee of good ageing in terms of health" https://doi.org/10.3390/bs12090331 "and that should be taken into account as a measure of health protection and improvement of functional skills that result in a better quality of life of older people and better physical health" https://doi. org/10.3390/socsci11060265 "being necessary to take into account among the relevant elements for older people in quality of life, health, social relations and staying active, physical activity being one of the most important disease prevention and health promotion strategies on a physical and cognitive scale" https://www.mdpi.com/2254-9625/7/3/135

Towards the end of your discussion I suggest you incorporate a few paragraphs in response:

(a) what theoretical implications does this work have for scientists reading this work, for theorists in the field or colleagues?

b) strengths of your work in relation to other studies.

c) What practical implications does this work have for older people?

d) future line of research arising from this work that needs to be covered.

I hope you get better visibility this way! Congratulations, I loved your work!

My sincere congratulations for the work.

Author Response

Dear reviewer,

Please see the attachment,

Best Regards,

Noelia

Reviewer 3 Report

Overall, this article is interesting and from a methodological standpoint well designed. However, I have considerable doubts about many of its elements. It is certainly written without pietism, there are many typos or carelessness in it, and it is not prepared according to the journal’s guidelines. Of the greater doubts, I have to mention the data analysis, which is correct to some extent, although I am not convinced that it fully reflects the article's assumptions. Below is a thorough review of major and minor issues. Certainly, this article still requires much work before publication.

Major issues:

1.     Move “2.4. Procedure information” to “2.1. Participants”.

2.     Line 201-203 – “The evaluation was carried out in these centers.” What centers? There is information regarding three sessions. What session do you referee? Session of data collection? This should be clarified. 

3.     Add information regarding, if was the selection targeted and when was the survey conducted.

4.     For the purpose of the study, the authors wrote that they included people without dementia, but the exclusion criterion does not include this information. If the authors wrote this in the context of the MMSE, the handle should have been changed to cognitive impairment (CI). On the other hand, if you excluded subjects with any clinically diagnosed dementia it should be described in the exclusion criteria.

5.     Outcomes measures should be described in a little more detail. Cronbach’s alpha for each of the tests used should also be given (like you did with EAE.

6.     Why was it decided to include 63 people, was a sample size calculation conducted?

7.     I am puzzled by the rationale of using hierarchical regression with one variable in one block in this article. I am not convinced of such data analysis. The advantage of hierarchical regression is that you can create your own model by including blocks on variables that are connected in some way. This advantage you have not used here, because your model is simply throwing in more variables. In your case, it would be to include age and number of years of education in one block, and IA and IP EAE and cortisol levels in another (as they examine the same thing in a different way). In contrast, if the authors wanted to stay with individual variable testing, then it would be better to use stepwise regression in my opinion.

8.     Change “3. Analysis” into “2.4. Data analysis” and put it into the methods section. Lines 221-224 – correct them and put at the beginning of data analysis. 

9.     In this article we have two main elements, i.e. cognitive performance (many variables) and stress (3 variables). And for some reason, the authors also included another variable on coping mechanism (resilience), whose presence in my opinion is not justified here. Of course, in a regression, everything depends on the model created, and reading the article I get the impression that the authors wanted to determine the roles of threats (hence so many stress variables) rather than the roles of protective factors. To determine the role of protective factors it would have been better to use mediation analysis to see the indirect effect of protective factors. Or use SEM or path analysis. Please note that by throwing too many different variables into the hierarchical regression model, the interpretation of R2 is hampered. In my opinion, it is not reasonable to include resilience in this regression model.

10.  Line 277-279 – “Psychological resilience stands out as the best predictor of overall cognitive performance, as it contributes to relevant domains such as working memory and verbal 278 fluency.” Is this supported by the results? Age and education seem to be better predictor. 

11.  I am not convinced to write about inverse association. First, inverse means "opposite or contrary in position, direction, order, or effect." Regression don't assess that. Second, in your results, in only one variable was past stress a significant predictor. You see no pattern for IP-EAE, and with such a small number of included participants, generalizing an individual result is not advisable.

Minor issues:

1.     Prepare references according to the journal requirements.

2.     Correct authors names.

3.     Do not write outcomes with hyphens.

4.     When you are describing outcome, provide scale acronym in brackets. Example Stressful Events in the Elderly Scale (EAE)

5.     Don’t provide sentences like “[known as Escala de Acontecimientos 179 Estresantes en Ancianos in Spanish]”. Instead simply write in the description, that this is a Spanish scale. 

6.     Delete dot after “2.3. Stress testing.”

7.     Line 146 – expand the MMSE acronym here, but delete al 2.2.1. point

8.     Table 1 – add information regarding MMSE

9.     Table 1 – change “min-max” to “range”

10.  Add the statements required by the journal

11.  Table 2 - Correct IA-SEA

Author Response

(The authors gave the same response as above.)

Round 2

Reviewer 1 Report

I think most of the changes suggested were adequately resolved. I only have some doubts about the sample size and how it was calculated. I do not even know if it would be better not to specify how it was done. I think it is not right to calculate it based on a single variable.

Author Response

Dear reviewer, 
Thank you very much for your comment and recommendation to improve the article.
We thank you for your assessment of the change included in the new version of the manuscript. We understand your doubts about the sample size calculation. The study by Pulopulos et al. (2014) helped us to have a baseline estimate but we considered it to be small and decided to increase the estimated size as we had only one variable in common. However, it was not possible to have any other study with which to carry out another type of estimate and reviewer number 3 has expressly asked us to indicate the calculation made. For this reason, we ask you to bear in mind that eliminating the paragraph relating to the sample calculation would go directly against the suggestion of the third reviewer and this would mean a penalty in the assessment that he/she will make of the manuscript

Best regards,

Noelia Saez-Sanz
